# Fully Integrated 3D-Printed Electronic Device for the On-Field Determination of Antipsychotic Drug Quetiapine

**DOI:** 10.3390/s21144753

**Published:** 2021-07-12

**Authors:** Katerina Ragazou, Rallis Lougkovois, Vassiliki Katseli, Christos Kokkinos

**Affiliations:** Laboratory of Analytical Chemistry, Department of Chemistry, National and Kapodistrian University of Athens, 157 71 Athens, Greece; ragazoukat@gmail.com (K.R.); rloug.96@gmail.com (R.L.); lilikats0@gmail.com (V.K.)

**Keywords:** 3D-printing, sensor, voltammetry, quetiapine

## Abstract

In this work, we developed a novel all-3D-printed device for the simple determination of quetiapine fumarate (QF) via voltammetric mode. The device was printed through a one-step process by a dual-extruder 3D printer and it features three thermoplastic electrodes (printed from a carbon black-loaded polylactic acid (PLA)) and an electrode holder printed from a non-conductive PLA filament. The integrated 3D-printed device can be printed on-field and it qualifies as a ready-to-use sensor, since it does not require any post-treatment (i.e., modification or activation) before use. The electrochemical parameters, which affect the performance of the sensor in QF determination, were optimized and, under the selected conditions, the quantification of QF was carried out in the concentration range of 5 × 10^−7^–80 × 10^−7^ mol × L^−1^. The limit of detection was 2 × 10^−9^ mol × L^−1^, which is lower than that of existing electrochemical QF sensors. The within-device and between-device reproducibility was 4.3% and 6.2% (at 50 × 10^−7^ mol × L^−1^ QF level), respectively, demonstrating the satisfactory operational and fabrication reproducibility of the device. Finally, the device was successfully applied for the determination of QF in pharmaceutical tablets and in human urine, justifying its suitability for routine and on-site analysis.

## 1. Introduction

Quetiapine fumarate (QF) is an atypical antipsychotic drug with a dibenzoethiazepine structure, chemically known as 2-[2-(4-dibenzo[b,f][1,4]thiazepin-11-yl-1-piperazinyl) ethoxy] ethanol hemifumarate. QF serves as antagonist of various neurotransmitters, such as serotonin and norepinephrine and it is used for the treatment of schizophrenia, while it is the monotherapeutic agent for the bipolar I and bipolar II disorders [1]. Since QF is a top-selling drug, accurate and sensitive analytical methods are required for the evaluation of its quantity in pharmaceutical products and in biological fluids. Several analytical methods have been applied for the determination of QF concentration, including HPLC combined with mass spectrometry and UV detection, as well as spectrophotometric and capillary zone electrophoretic methods [2,3,4,5]. However, these methods require expensive and central laboratory facilities operated by experienced personnel, are time-consuming, and involve tedious sample preparations; thus, they cannot be used for on-site analyses.

Alternatively, electrochemical methods present analogous sensitivity to the above methods but using cheaper and portable instrumentation, more rapid and simpler assays and thus, they are far more fit for on-field screenings [6,7]. Moreover, the research field of electronic sensors is quickly developing through the ongoing application of cutting-edge technologies [8,9]. Nevertheless, only a few works on the electrochemical determination of QF have been proposed in the literature so far, using bare or modified (with molecular imprinted polymers, graphene, and selective films) glassy carbon and carbon paste electrodes [10,11,12,13,14,15,16,17,18]. All these electrochemical methods of QF determination involve the use of separate “large-size” external reference and counter electrodes; therefore, they do not have any degree of miniaturization and integration. On the other hand, three-dimensional (3D) printing can accomplish the production of completed electronic units. Fused deposition modeling (FDM) is an innovative 3D printing process in which a sensor is CAD designed and then printed from thermoplastic filaments by a 3D printer. This digital process requires low-cost and desktop-sized printers and it provides ease of operation by non-trained handlers, design flexibility and transferability via e-mail, while it produces eco-friendly and disposable sensors [19,20,21,22,23,24,25,26,27,28,29].

Here, exploiting the advantages of 3D printing technology, we introduce a miniature all-3D-printed and fully integrated device for the voltammetric determination of the antipsychotic drug QF. The device is entirely printed by a dual-extruder 3D printer and is composed of three electrodes printed from a carbon black-loaded polylactic acid (PLA) conductive filament and an electrode holder printed from a PLA non-conductive filament (Figure 1). The presented 3D-printed device qualifies as a ready-to-use sensor, since it integrates the working, counter, and reference electrodes (WE, CE, RE, respectively) and does not require any modification or activation step before use for the sensitive and selective determination of QF.

## 2. Materials and Methods

### 2.1. Reagents and Apparatus

The white color non-conductive filament, with diameter of 1.75 mm, was polylactic acid (PLA) from 3DEdge (Athens, Greece). Two carbon black-loaded conductive filaments (all with diameters of 1.75 mm) were examined: (i) acrylonitrile butadiene styrene (ABS) from 3DEdge and (ii) PLA from Proto-Pasta (Vancouver, BC, Canada). The pharmaceutical tablets (commercial name: Seroquel from Astra Zeneca (Cambridge, UK)) contained on average 25 mg of QF and were obtained from a local drug store. The pharmaceutical tablets contained as excipients the following materials: povidone, calcium hydrogen phosphate dihydrate, microcrystalline cellulose, sodium starch glycollate Type A, lactose monohydrate, magnesium stearate, hypromellose, macrogol, titanium dioxide, and iron oxide red. All the other reagents were from Merck (Athens, Greece). The Britton–Robinson buffer (BRB) was prepared by mixing phosphoric acid, acetic acid, and boric acid, while 0.1 mol × L^−1^ solution of NaOH was used to adjust the pH. The phosphate buffer (PB) was prepared by mixing Na_2_HPO_4_ and NaH_2_PO_4_ and the pH value was adjusted with 0.1 mol × L^−1^ solution of HCl or NaOH. The voltammetric measurements were carried out with the portable EmStat3 potentiostat (Palm Sens (Houten, The Netherlands)) and data treatment was carried out with the PS Trace 4.2 software (Palm Sens). The HPLC system was from Waters (Milford, MA, USA) and consisted of a 2695 binary HPLC LC pump, a 2487 UV-Visible dual-absorbance detector, and a reverse-phase C18 column (75 mm × 4.6 mm I.D., particle size 3.5 µm).

### 2.2. Fabrication of the 3D Printed Device 

The integrated three-electrode device was designed with the Tinkercad software (https://www.tinkercad.com accessed on 12 July 2021) and then printed by the Creator Pro dual-extruder 3D printer from Flashforge. The printing conditions were set to 60 °C for the platform and 200 and 220 °C for the head dispensers for PLA and ABS, respectively, using the Flashprint software (https://www.flashforge.com/download-center accessed on 12 July 2021). The dimensions and a photograph of the 3D-printed device are illustrated in Figure 1.

### 2.3. Voltammetric Measurements and Samples Analysis

The connection of the 3D-printed device to the potentiostat was accomplished with crocodile clips. For the square wave voltammetric (SWV) measurements of QF, the solution was stirred at 1000 rpm for 120 s at −0.3 V and the voltammetric scan from 0.5 to 1.5 V was operated in static solution. For the SWV analysis of pharmaceutical tablets, 3 tablets of Seroquel (25 mg QF per tablet) were pulverized and dissolved in 100 mL of double-distilled water. Then, 10 µL of the resulting solution was transferred in the electrochemical cell, which was previously filled with 10 mL of 0.1 mol × L^−1^ acetate buffer (pH 4.8). The concentration of QF in the Seroquel tablets was also measured with HPLC-UV using a reverse-phase C18 column in a mobile phase of 40:60 *v*/*v* phosphate buffer (pH 3.0) and acetonitrile, a flow rate of 0.8 mL × min^−1^, and detection at 291 nm. The urine sample was obtained by a healthy 24-year-old female volunteer, within her informed written consent and the measurements were carried out in accordance with GCP regulations. The urine sample was spiked with QF to a final concentration of 3 × 10^−6^ mol × L^−1^ (which normally is found in the urine after treatment with a daily dose of 50 mg QF) [15] and then diluted 1:5 with 0.1 mol × L^−1^ of acetate buffer (pH 4.8). For the analysis of the Seroquel tablets and urine sample, the standard addition method was applied for the quantification of QF. All potentials at the 3D printed device are referred with respect to the carbon black-loaded PLA reference electrode.

## 3. Results and Discussion

### 3.1. Voltammetric Determination of Quetiapine Fumarate

Initially, the electrochemical behavior of QF was examined through cyclic voltammetry (CV) in 0.1 mol × L^−1^ of acetate buffer (pH 4.8) containing QF at two different concentrations. As depicted in Figure 2A, the oxidation of QF at the 3D-printed device presented the only anodic peak at about 1.0 V, which increased with the increase in QF concentration. On the cathodic reverse scan, there was an absence of the reduction peak, indicating the irreversible nature of the reaction at the 3D-printed device carbon black-loaded PLA WE. The QF oxidation reaction is most likely related to the aliphatic nitrogen of the piperazine ring, and a possible reaction of QF on the carbon black-loaded PLA WE is presented in Figure 2B [14].

The thermoplastic filament material used for printing the WE of the device affected the voltammetric determination sensitivity of QF, and for this reason two different carbon black-loaded conductive filaments (PLA and ABS) were examined in terms of the SWV oxidation peak heights of QF. The WE printed from carbon black-loaded PLA filament exhibited higher responses and therefore was selected for the printing of the WE, CE, and RE (Figure 3A). The differences in electrochemical responses between the two tested carbon black-loaded filaments can be attributed to their different compositions (proprietary information) and specifically to their carbon-black content, which affects their resistivity (resistivity of 0.1 Ω·cm for the conductive ABS and 15 Ω·cm for the conductive PLA).

The performance of the integrated 3D-printed carbon black-loaded RE in terms of potential stability and between-electrode potential reproducibility is a core operational feature of the device and was thoroughly examined. In particular, the potential stability of the 3D printed RE was tested for 20 repetitive SWV measurements, while the within RE potential reproducibility was estimated via the comparison of the responses obtained by 8 different 3D printed REs. Both examinations were conducted in a solution containing 50 × 10^−7^ mol × L^−1^ QF in 0.1 mol × L^−1^ acetate buffer (pH 4.8). The potential, where the QF oxidation peak appears, remained statistically constant during the 20 repetitive measurements (Figure 3B), while the % relative standard deviation (% RSD) of the appeared peak potential of QF at the 8 different printed REs was 3.8%, demonstrating acceptable potential stability and reproducibility of the printed carbon black-loaded PLA RE.

To enhance the electrochemical performance of the 3D-printed integrated sensor in the voltammetric determination of QF, the composition of the supporting electrolyte, the effect of the accumulation time and potential, and the scanning waveform were optimized in a solution containing 50 × 10^−7^ mol × L^−1^ QF. QF is a weak base with 3.3 and 6.8 pKa, and in solutions with acidic pH values the piperazine moiety is protonated, increasing the adsorption of positively charged molecules of QF on the electrode surface [12,13,14,15,16]. It has been shown before that, at neutral and alkaline pH values, the peak heights of QF are reduced and may be split into two peaks [12,13,14,15,16]. The applied potential of the 3D-printed device after optimization was −0.3 V (Figure 3C), facilitating the adsorption of positively charged QF at acidic pH. Therefore, different supporting electrolytes with acidic pH values were examined (0.1 mol × L^−1^ acetate buffer pH 4.8, 0.1 mol × L^−1^ PB pH 4.5, Britton–Robinson pH 4, HCl pH 3) and the best responses, in terms of peak shape and height of QF, were obtained in 0.1 mol × L^−1^ acetate buffer (pH 4.8) and in 0.1 mol × L^−1^ PB pH 4.5 (Figure 3D). Acetate buffer (0.1 mol × L^−1^, pH 4.8) was selected as the supporting electrolyte for the rest of the experiments. The influence of the accumulation time on the QF voltammetric responses was examined in the range 0–360 s in a solution containing 50 × 10^−7^ mol × L^−1^ QF in 0.1 mol × L^−1^ acetate buffer (pH 4.8). As depicted in Figure 3E, at low accumulation times (up to 120 s) the oxidation peak current of QF increased rapidly with the accumulation time, while at higher accumulation periods the peak current continued to increase but at a slower rate. For the further experiment, an accumulation time of 120 s at 1000 rpm was selected, combining good sensitivity and short voltammetric measurements. Additionally, differential pulse (DP) and SW waveforms were compared (Figure 3F). The DP mode produced a sloping baseline and low sensitivity. The SW mode presented a higher sensitivity with a flatter baseline and thus was chosen for the experiments. Finally, the SW parameters (SW frequency (in the range 12.5–100 Hz), SW step increment (in the range 1–16 mV), SW pulse height (in the range 10–80 mV)) and the SWV response of QF were optimized in a solution containing 50 × 10^−7^ mol × L^−1^ QF. The best compromise between the sensitivity, duration of the measurements, peak shape, and background characteristics was achieved at a frequency of 50 Hz, step increment of 4 mV, pulse height of 40 mV, accumulation time of 120 s, and applied potential of −0.3 V.

Calibration for QF at various concentrations was carried out in 0.1 mol × L^−1^ acetate buffer (pH 4.8) with the 3D-printed device after 120 s accumulation time of QF on the WE surface, which was polarized at −0.3 V. The oxidation peak presented a linear concentration dependence in the tested concentration range (5 × 10^−7^–80 × 10^−7^ mol × L^−1^ for QF) with R^2^ = 0.997 (Figure 4). The limit of detection (LOD) was 2 *×* 10^−9^ mol × L^−1^ and estimated through the equation LOD = 3 sd/a (where sd is the standard deviation of the intercept of the calibration plot and a is the slope of the calibration plot). As presented in Table 1, the LOD of the proposed method is lower than that offered by previously reported QF electronic modified sensors [10,11,12,13,14,15,16]. The within-device reproducibility was calculated by measuring the responses of 50 × 10^−7^ mol × L^−1^ QF via eight repeated measurements and the % relative standard deviation (RSD%) was 4.3%. The between-device reproducibility (in terms of the % RSD for six different devices) was 6.2% at the 50 × 10^−7^ mol × L^−1^ QF level. These electroanalytical results demonstrate the satisfactory operational and fabrication reproducibility of the 3D-printed device.

The effect of some possible interfering substances, which can be present in pharmaceutical and biological samples, was studied by the addition of the compounds to a solution containing 50 × 10^−7^ QF. Glucose, fructose, saccharose, urea, ascorbic acid, citric acid, Pb(II), Zn(II), Cd(II), or Cu(II) did not interfere at a concentration ratio of 1:10 (standard solution: interference compound), indicating the good selectivity of the presented electrochemical method and its practicality for QF determination in biological samples.

### 3.2. Applications

In order to assess the analytical usefulness of the presented electrochemical method, the integrated 3D-printed device was applied for the quantification of QF in urine and commercially available Seroquel tablets. Both samples were treated following the procedures described in Section 2.3. The standard addition method was used for both sample analyses calculating the respective recovery values. The quantity of QF in pharmaceutical tablets was also measured with HPLC-UV and the determined value was 25.1 ± 0.4 mg per tablet. Satisfactory recoveries for QF were obtained confirming the accuracy of the presented electrochemical method and its suitability for routine on-field analyses. In particular, in the case of the pharmaceutical tablets the determined quantity was (24.8 ± 0.6 mg per tablet), giving a recovery of 99 ± 3%, while in the case of the spiked urine sample the obtained recovery was 103 ± 5% (Figure 5).

## 4. Conclusions

This work describes the voltammetric determination of trace QF in pharmaceutical tablets and in human urine using an all-3D-printed three-electrode device. The electrochemical device exploits the important advantages of 3D printing technology in terms of fabrication speed, reproducibility, affordability, and eco-friendliness, while the presented electrochemical method is simple and fast, eliminating any requirement for sample pretreatment or separation steps. Therefore, this e-transferable and ready-to-use sensor is a promising candidate for the routine and on-site analysis of QF.

## Figures and Tables

**Figure 1 sensors-21-04753-f001:**
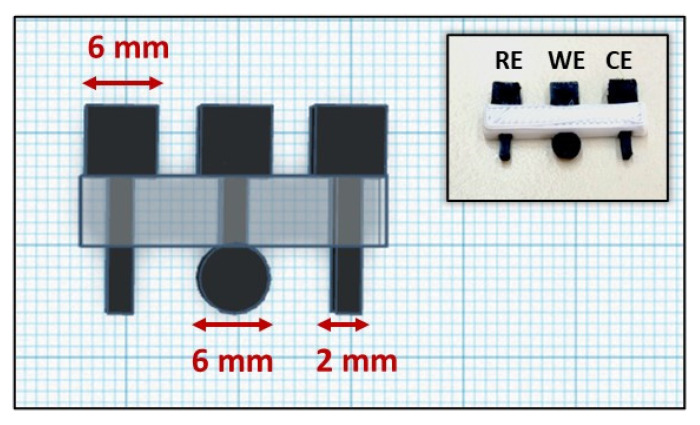
Schematic illustration and photograph of the 3D-printed device.

**Figure 2 sensors-21-04753-f002:**
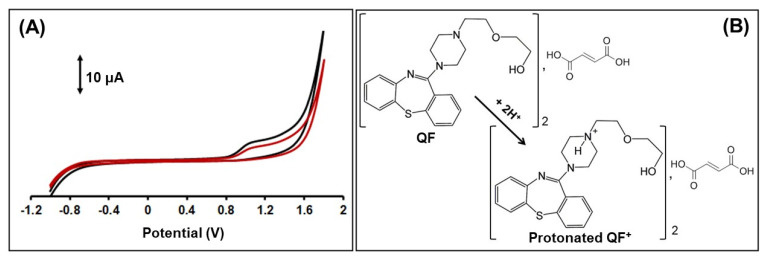
(**A**) CV responses of the 3D-printed device in the presence of 50 × 10^−7^ mol × L^−1^ (red line) and 200 × 10^−7^ mol × L^−1^ QF. Scan rate: scan rate, 50 mV × s^−1^; electrolyte, 0.1 mol × L^−1^ acetate buffer (pH 4.8). (**B**) Possible reaction of QF on the carbon black-loaded PLA WE.

**Figure 3 sensors-21-04753-f003:**
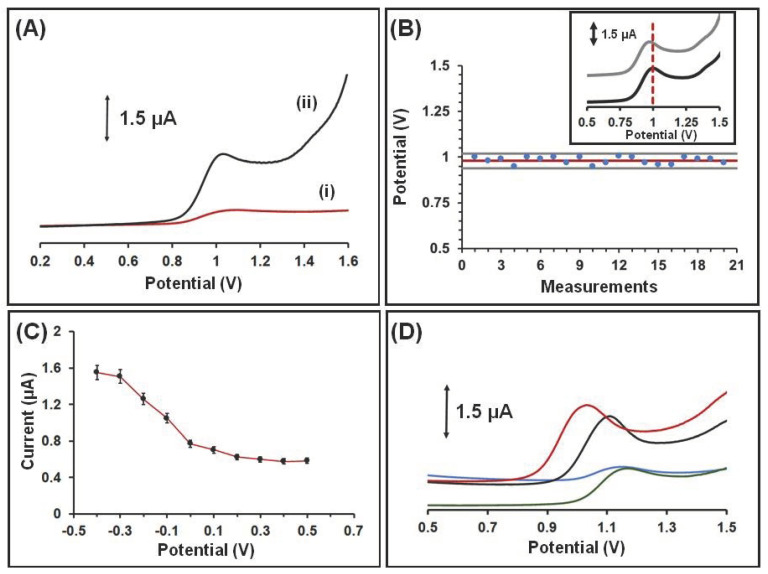
(**A**) SWV responses of 50 × 10^−7^ mol × L^−1^ QF in the 3D-printed device using carbon black-loaded WEs printed from: (i) ABS and (ii) PLA. (**B**) The stability of the 3D-printed reference electrode potential. The measurements were conducted in a solution containing 50 × 10^−7^ mol × L^−1^ QF in 0.1 mol × L^−1^ acetate buffer (pH 4.8). The red line is the mean value of the 20 measurements and grey lines define an interval equal to mean value ± 2 × sd, where sd is the standard deviation of the 20 measurements. Inset: representative voltammograms of 50 × 10^−7^ mol × L^−1^ QF from a series of 2 consecutive SW scans: (black line) 1st and (grey line) 20st scan. (**C**) The effect of the applied potential on the SWV response of 50 × 10^−7^ mol × L^−1^ QF. Error bars are the mean value ± sd (*n* = 3). (**D**) Comparative SW voltammograms of 50 × 10^−7^ mol × L^−1^ QF in: (red line) 0.1 mol × L^−1^ acetate buffer pH 4.8, (black line) 0.1 mol × L^−1^ phosphate buffer pH 4.5, (green line) Britton–Robinson pH 4, (blue line) HCl pH 3. (**E**) The effect of accumulation time on the SWV response of 50 × 10^−7^ mol × L^−1^ QF. Error bars are the mean value ± sd (*n* = 3). (**F**) The effect of the scanning waveform on the voltammetric response of 50 × 10^−7^ mol × L^−1^ QF: (i) SW, (ii) DP.

**Figure 4 sensors-21-04753-f004:**
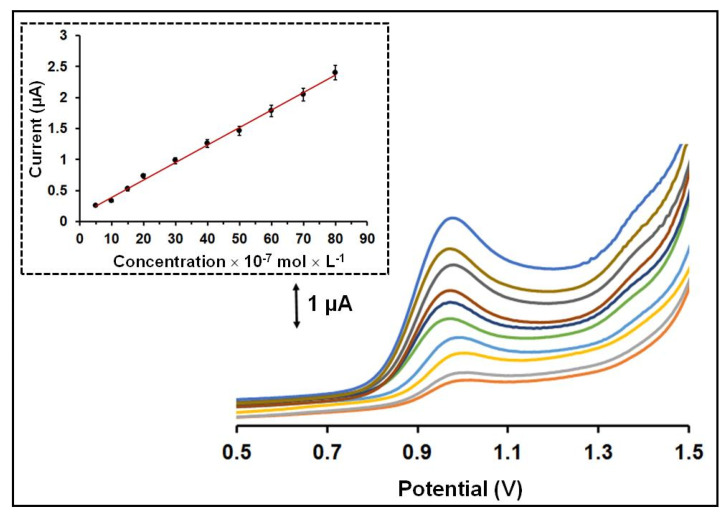
SW voltammograms obtained at the 3D-printed integrated sensor for QF in the concentration range 5 × 10^−7^–80 × 10^−7^ mol × L^−1^ in 0.1 mol × L^−1^ acetate buffer (pH 4.8) after accumulation for 120 s. The respective calibration plot for QF is shown as inset. The points in the calibration plot are the mean value ± sd (*n* = 3). Applied potential −0.3 V, accumulation time 120 s.

**Figure 5 sensors-21-04753-f005:**
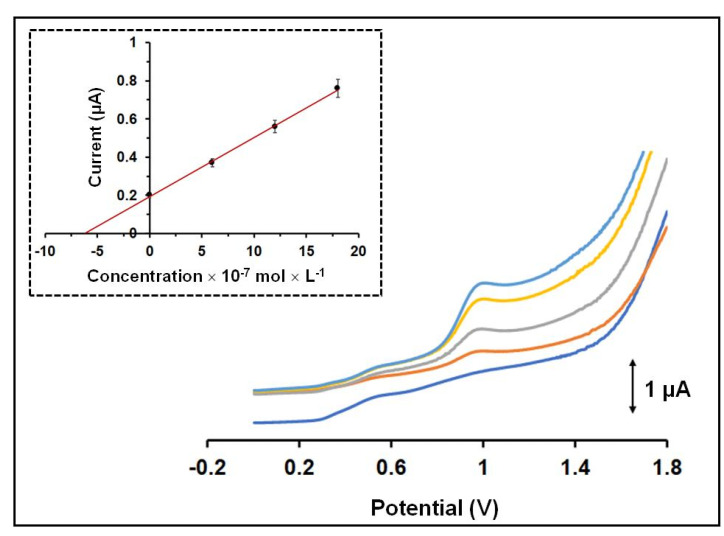
SW voltammograms for the determination of QF in urine sample using the 3D-printed sensor. Traces from below: unspiked urine sample, urine sample spiked with QF and 3 standard additions of 6 × 10^−7^ mol × L^−1^ QF. The respective standard additions plot for QF is shown as inset. The points in the plot are the mean value ± sd (*n* = 3). Applied potential: −0.3 V; accumulation time: 120 s.

**Table 1 sensors-21-04753-t001:** Comparison of the 3D-printed device with other QF electrochemical sensors, in terms of linear range and LOD.

Sensor	Studied Linear Range (mol × L^−1^)	LOD (mol × L^−1^)	Reference
CPE modified with molecularly imprinted polymer	1.6 × 10^−8^–2.0 × 10^−5^	5 × 10^−9^	10
GCE	2 × 10^−8^–5 × 10^−6^	1 × 10^−8^	11
GCE modified with polymeric film	8.0 × 10^−8^–7.5 × 10^−6^	1.9 × 10^−8^	12
PVC membrane sensor	1 × 10^−7^–2 × 10^−2^	2 × 10^−7^	13
GCE modified with carbon black nanoparticles	5.0 × 10^−8^–3.5 × 10^−6^	7 × 10^−9^	14
GCE modified with graphene nanoplatelets	1 × 10^−7^–1 × 10^−5^	2.2 × 10^−8^	15
GCE	4 × 10^−6^–2 × 10^−4^	4 × 10^−8^	16
Dropping mercury electrode.	2.1 × 10^−5^–1.2 × 10^−4^	1.6 × 10^−7^	17
PVC membrane sensor	1 × 10^−5^–2 × 10^−2^	3.2 × 10^−6^	18
3D printed carbon black loaded PLA	5 × 10^−7^–8 × 10^−6^	2 × 10^−9^	This work

CPE: carbon paste electrode; GCE: glassy carbon electrode; PVC: polyvinyl chloride; PLA: polylactic acid.

## Data Availability

Not applicable.

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
