# Peer review of "Fully Integrated 3D-Printed Electronic Device for the On-Field Determination of Antipsychotic Drug Quetiapine"

_sensors, 2021, doi:10.3390/s21144753_

Round 1
Reviewer 1 Report
They fabricated 3D printed QF electrochemical sensors by using carbon-loaded PLA. This manuscript has interesting points, so it may be acceptable for this journal after several correction.
More detail information about 3D printer filament material should be added in this manuscript in the point reproductivity.
Regarding Table 1, the digit should be adjusted.
I think the validation of drug concentration by using other method is necessary to confirm the usefulness of this method.
Reviewer 2 Report
This is a nice contribution of Kokkinos group on 3D printed electrochemical sensors that deserves publication in the journal after a few corrections as follow:
- Introduction: I suggest including this important review on 3D printed electrochemical sensors https://doi.org/10.1016/j.aca.2020.03.028
- Considering the focus of the paper on the determination of an important species in pharmaceutical and specially in urine samples, I suggest including more references that performed the analysis of similar samples using 3D printed electrochemical sensors.
- Have the authors evaluated a surface treatment of the working electrodes? Was surface polishing performed? Thare are many papers showing the improvement on electrochemical or chemical surface treatment of Protopasta carbon black PLA electrodes.
- Figure 2: the authors mentioned that difference of the voltammetric behavior may be due to the amount of carbon content. What is the amount of carbon in each electrode?
- Could the authors include a possible (tentative) mechanism for the electrochemical oxidation based on previous reports? It would be nice to see the chemical structure of the analyte.
- Figure 3: I suggest include the first and last (20th) scans in (A) regarding the repeatability test.
- Phosphate buffer pH 6 showed similar current as acetate buffer with the advantage of a lower oxidation potential. Is there another reason to select acetate buffer?
- Accumulation time: What is the stirring rate in rpm ? Do error bars in B correspond to n=3?
- How was the selection of the deposition potential? What about SW parameters?
- Please include deviation for recovery values of the analyzed samples.
- Please include information on bioethical certification for urine collection and analysis. I am not sure if the journal requires such a certification.
- Caption of Figure 5 is not clear. Is the blue line the pure sample? Please correct it.
Reviewer 3 Report
The authors have presented an interesting subject with clinical interest. The electrochemical sensors have found many valuable applications in different analytical purposes in pharmaceutical analysis.
The manuscript has been well prepared, based on a recent but not extensive literature database. However, so it seems that the idea is not new, and this paper does not impressive deals with this subject [1, 2]. The article has scientific soundness and introduces the new method for the detection of quetiapine, but there are some less or more important aspects that authors should improve.
Generally, some parts of the manuscript are hard to read with logical, stylistically and typographical errors. It seems that describing new sensors additional studies should be carried out such as scanning electron microscopy (SEM) and electrochemical impedance spectroscopy (EIS) for better sensor’s characterization. Moreover, the application of quasi-reference electrode for new sensor introduces high variability of the measuring equipment.
Some of the suggested corrections are listed below:
- Firstly, the method used is called voltammetry (voltammetric method).
- Some of the sentences are unclear and should be rewritten or corrected: Lines 38-40, 59-62, 107-110…
- There are many places in the text with a lack of super or subscripts.
- Pharmaceutical product contains concrete API content strictly regulated by local Pharmacopoeia, which means the API content is not average.
- The authors should clarify why they chose SWV voltammetry with selected parameters, giving no satisfactory reason for not applying i.e. DP voltammetry. I suppose this was not checked properly. Normally, I expect voltammograms with well-shaped, sharp peak(s) or any kind of signal suitable for interpretation. The obtained curves have formed quasi-peaks with not possible FWHM measurements, particularly for the urine sample.
- R2 is not the same as the correlation coefficient.
- Fast voltammetric measurements using 120 s of accumulation time, is that real?
- The study on "healthy volunteer" is regulated by the proper Bioethics Committee. It was not mentioned in the Acknowledgement section. Moreover, study on psychoactive substance (quetiapine) is strictly regulated in some countries.
- References section should be carefully checked and corrected according to Sensors style or Reference Manager.
Finally, In my opinion, the manuscript is unsuitable for publication in its current form. Below, I suggest a manuscript with the well-conducted study:
- Yang C, Cao Q, Puthongkham P, Lee ST, Ganesana M, Lavrik NV, Venton BJ. 3D-Printed Carbon Electrodes for Neurotransmitter Detection. Angew Chem Int Ed Engl. 2018 Oct 22;57(43):14255-14259. https://doi.org/10.1002/anie.201809992
- Vaněčková E, Bouša M., Nováková Lachmanová Š, Rathouský J, Gál M, Sebechlebská T, Kolivoška V, 3D printed polylactic acid/carbon black electrodes with nearly ideal electrochemical behaviour, Journal of Electroanalytical Chemistry, Volume 857, 2020, 113745, https://doi.org/10.1016/j.jelechem.2019.113745
Round 2
Reviewer 3 Report
Dear Authors,
Thank you for the incorporation of my suggestions, all my concerns have been addressed. There are still some aspects that could be corrected but without them, the manuscript does not lose its attractiveness and sound scientific good (as communication from).
I suggest some corrections needed:
- line 173: "time of measurement" should be changed "ie. analysis time, duration of the measurements"
- Figure 3 caption: "The stability of the 3D printed reference electrode potential" looks better
Ad vocem to the responses:
SEM and EIS are techniques utilized in order to know the characteristics of an electroanalytical cell (particularly newly presented), not only WE but RE also. Application of solely CV is acceptable in this work (communication form), but in full article is a condition sine qua non (absolutely necessary).
I have suggested the articles as an example of the gold standard (not the subject) of a well-conducted study, in my humble opinion.
Author Response
Response to Reviewer 3
1) line 173: "time of measurement" should be changed "ie. analysis time, duration of the measurements".
Response
The suggestion has been adopted in the revised manuscript.
2) Figure 3 caption: "The stability of the 3D printed reference electrode potential" looks better
Response
The suggestion has been adopted in the revised manuscript.